

# Modelling approaches for atmospheric ion-dipole collisions: all-atom trajectory simulations and central field methods

Ivo Neefjes[1], Roope Halonen[2], Hanna Vehkamäki[1], and Bernhard Reischl[1]

[1]Institute for Atmospheric and Earth System Research / Physics, Faculty of Science, University of Helsinki, P.O. Box 64, FI-00014, Finland
[2]Center for Joint Quantum Studies and Department of Physics, School of Science, Tianjin University, 92 Weijin Road, Tianjin 300072, China

**Correspondence:** Ivo Neefjes (ivo.neefjes@helsinki.fi), Roope Halonen (roope@tju.edu.cn)

**Abstract.** Ion-dipole collisions can facilitate the formation of atmospheric aerosol particles, and play an important role in their detection in chemical ionization mass spectrometers. Conventionally, analytical models, or simple parametrizations, have been used to calculate rate coefficients of ion-dipole collisions in the gas phase. Such models, however, neglect the atomistic structure and charge distribution of the collision partners. To determine the accuracy and applicability of these approaches at atmospheric

conditions, we calculated collision cross sections and rate coefficients from all-atom molecular dynamics collision trajectories, sampling the relevant range of impact parameters and relative velocities, and from a central field model using an effective attractive interaction fitted to the long-range potential of mean force between the collision partners. We considered collisions between various atmospherically relevant molecular ions and dipoles, as well as charged and neutral dipolar clusters. Based on the good agreement between collision cross sections and rate coefficients obtained from molecular dynamics trajectories and a

generalized central field model, we conclude that the effective interactions between the collision partners are isotropic to a high degree, and the model is able to capture the relevant physico-chemical properties of the systems. In addition, when the potential of mean force is recalculated at the respective temperatures, the central field model exhibits the correct temperature dependence of the collision process. The classical parametrization by Su and Chesnavich [J. Chem. Phys., 76, 5183–5185, 1982], which combines a central field model with simplified trajectory simulations, is able to predict the collision rate coefficients and their

temperature dependence quite well for molecular systems, but the agreement worsens for systems containing clusters. Based on our results, we propose the combination of potential of mean force calculation and central field model as a viable and elegant alternative to brute force sampling of individual collision trajectories over a large range of impact parameters and relative velocities.

## 1 Introduction

In the atmosphere, gas-phase molecules can aggregate to form molecular clusters, and subsequently grow into larger sized atmospheric aerosol particles, in a process called new particle formation (NPF) (Gordon et al., 2017). Once formed, atmospheric aerosol particles affect the global climate both directly, by scattering and absorbing solar radiation, and indirectly, by acting as nuclei for the formation of clouds (Kurtén et al., 2003). Aerosol particles are, furthermore, responsible for adverse



health effects through air pollution (Falcon-Rodriguez et al., 2016). It is estimated that the majority of atmospheric aerosol

particles originates from NPF (Gordon et al., 2017; Yu and Luo, 2009). NPF is mainly driven by neutral pathways, involving trace gas molecules such as sulfuric acid and various bases. The presence of atmospheric ions can, however, significantly enhance NPF (Kirkby et al., 2016; Wagner et al., 2017). Atmospheric ions are formed under the influence of galactic cosmic rays and terrestrial radioactivity (Zhang et al., 2011), and stabilize newly formed atmospheric clusters. Ions, furthermore, play an important role in the detection and characterization of atmospheric clusters through chemical ionization mass spectrometry,

which depends on collisions between the studied atmospheric clusters and ions to form detectable charged clusters (Zhao et al., 2010).

The first stage of NPF is the gas-phase collision between single molecules or ions to form a dimer. For a theoretical description of NPF, it is therefore crucial to properly characterize the thermodynamics and kinetics of these initial collisions. In current NPF models, the cluster thermodynamics (e.g., cluster binding free energies and therefrom derived fragmentation rate

coefficients) are treated with high-level quantum chemical calculations (Elm, 2019; Elm et al., 2020), whereas the treatment of the cluster kinetics (e.g., collision cross sections and collision rate coefficients) is less sophisticated.

The theoretical prediction of collision kinetics is a long-standing topic throughout physics and chemistry (e.g., atmospheric chemistry, subatomic physics, and mass spectrometry), and thus several theoretical and computational methods have been developed. Collision rate coefficients generally depend on both the relative velocity between the collision partners and the fluid

density regime (Gopalakrishnan and Hogan Jr, 2011; Thajudeen et al., 2012). Here, we concentrate on methods developed for resolving canonical collision rate coefficients (i.e., the velocities of the collision partners follow the Maxwell-Boltzmann distribution) in the free molecular regime. An approximate estimate is obtained by assuming the collision partners to be non-interacting hard spheres with well-defined radii. Although intermolecular interactions are ignored in this approach, the hard-sphere model is widely used, especially for collisions between two neutral collision partners.

Neglecting the attractive forces between the collision partners can result in significant discrepancies with experiments, especially for systems with strong intermolecular interactions, such as systems containing ions. In 1905, Langevin (1905) derived an expression for the rate coefficient of a collision involving ion-neutral interactions using a central field approach. Although Langevin derived a compact equation specifically for the collision rate coefficients of systems with an ion-induced dipole interaction, later revisited by Gioumousis and Stevenson (Gioumousis and Stevenson, 1958), the central field approach

can be used with any attractive potential, e.g., for ion-"locked in" dipole (Moran and Hamill, 1963) and ion-averaged dipole orientation (Su and Bowers, 1973; Su et al., 1978) models.

In addition, various statistical models (often referred to as variational transition state theories), with quantized energy levels, exist for collision processes (Chesnavich et al., 1979, 1980; Troe, 1985, 1987; Clary, 1990; Georgievskii and Klippenstein, 2005). Interestingly, under equal assumptions, the statistical models give results identical to those of the central field models

(Chesnavich et al., 1979; Georgievskii and Klippenstein, 2005; Fernández-Ramos et al., 2006). One can also adopt a dynamical, rather than a statistical, approach: the collision cross sections and rate coefficients can be obtained by numerically solving the classical equations of motion with computational methods (Dugan Jr. and Magee, 1967; Chesnavich et al., 1980; Su and Chesnavich, 1982; Maergoiz et al., 1996a, b, c; Yang et al., 2018; Halonen et al., 2019; Goudeli et al., 2020). Based on trajectory





simulations between a point-like charged particle and a polar rigid rod, Su and Chesnavich (1982) obtained a parametrized
model for the collision rate coefficient of ion-dipole collisions. This model has been shown to give rather good results for
systems of small molecules and ions (Amelynck et al., 2005; Strekowski et al., 2019; Midey et al., 2001; Williams et al.,
2004; Woon and Herbst, 2009), and is widely used in atmospheric sciences (e.g., in the Atmospheric Cluster Dynamics Code
(McGrath et al., 2012)).

Although the aforementioned theoretical approaches are flexible and readily applicable, they often rely on simplified char-
acterizations of the studied collision system and the intermolecular interactions. This can potentially lead to significant inaccu-
racies in the predicted collision rate coefficients. As mentioned earlier, an ion-dipole complex is often reduced to a point-like
charge and a polar rod. However, especially for larger molecules (or clusters), the non-symmetric molecular structure and dy-
namic partial charge distribution should be considered to determine the actual strength of the interaction. Recently, Halonen
et al. (2019) compared collision rate coefficients obtained from the hard-sphere model and an atomistic molecular dynam-
ics (MD) model, including long range interaction, for a collision between two sulfuric acid molecules. The atomistic model
showed an enhancement of the collision rate coefficient by a factor 2.2. This enhancement factor is close to the discrepancy
between particle formation rates in experiments and a kinetic model reported by Kürten et al. (2014).

In this study, we examine collisions between one charged and one neutral dipolar collision partner. While there are typically
significantly less ions present compared to neutral molecules, ion-neutral collision rate coefficients are higher than for neutral-
neutral collisions due to relatively strong long-range interactions. Such collisions usually do not involve a significant electronic
activation energy barrier. However, the collision process does involve a *centrifugal barrier* due to the conservation of the
system's angular momentum which can lead to interesting, non-standard, temperature dependencies (Clary, 1990).

Here, as test systems, we considered collisions of the atmospherically relevant molecular dipole sulfuric acid ($H_2SO_4$),
with the anions bisulfate ($HSO_4^-$) and nitrate ($NO_3^-$), as well as the cations ammonium ($NH_4^+$) and dimethylammonium
(($CH_3)_2NH_2^+$). To study the effect of an increase in the size of the ion, we considered collisions between $H_2SO_4$ and the
sulfuric acid-bisulfate dimer ($[H_2SO_4 \cdot HSO_4^-]$). Likewise, to examine the effect of an increase in size of the dipole, we studied
collisions of the neutral bisulfate-dimethylammonium dimer with $HSO_4^-$ and $(CH_3)_2NH_2^+$. Lastly, we also looked at the
dimer-dimer collision between $[HSO_4^- \cdot (CH_3)_2NH_2^+]$ and $[H_2SO_4 \cdot HSO_4^-]$.

We carried out all-atom MD trajectory simulations of the collisions to determine the collision rate coefficient directly from
the collision probabilities in relevant ranges of the impact parameter and relative velocity. Additionally, we calculated the
potentials of mean force (PMF) between collision partners from well-tempered metadynamics simulations to determine the
*effective* potential, arising from the same underlying atomistic interactions, at finite temperature. Attractive interactions fitted
to the tail of the PMFs were used to predict collision cross sections and canonical rate coefficients using a central field model.
Lastly, we compared the analytical Langevin-Gioumousis-Stevenson model and the parametrization of Su and Chesnavich to
our robust atomistic MD results, and assessed the accuracy and applicability of those theoretical approaches.

The remainder of this paper is organized as follows: in Sect. 2 we present and discuss the different theoretical models, the
atomistic models of atmospherically relevant ions and dipoles studied, the PMF calculations, and the MD collision simulations.
In Sect. 3, we report and compare the results using the central field model based on the PMF, the MD trajectory simulations,



as well as the Langevin-Gioumousis-Stevenson model and Su and Chesnavich parametrization, for the same atomistic model
systems. In Sect. 4, we summarize our results and conclude the paper.

## 2  Theory and Methods

The formation of a molecular cluster through collisions requires asymptotic attractive intermolecular interaction potentials
which can be ideally modelled as a function of the distance $r$, separating the two collision partners:

$$U(r) = -A\left(\frac{r}{r_0}\right)^a,  \tag{1}$$

where $A$ is an interaction coefficient, $r_0$ is a distance parameter, and $a < 0$ is the characteristic interaction exponent. The
collision cross section and collision rate coefficient in an isotropic potential field given by Eq. (1) can be solved analytically
for collisions between point-like particles.

In the central field model, one of the collision partners is set to be stationary while the other approaches from infinitely far
away with some initial velocity $v_0$. The perpendicular distance between the initial trajectory and the center of the field is called
the impact parameter $b$. The initial configuration is illustrated in Fig. 1. As the collision partners form an isolated system, both
energy (initially only kinetic energy) and angular momentum, $\mu v_0 b$, are conserved, and the following equality holds during the
trajectory:

$$\frac{1}{2}\mu v_0^2 = U(r) + \frac{\mu v_0^2 b^2}{2r^2} + \frac{1}{2}\mu v^2.  \tag{2}$$

Here, $\mu$ is the reduced mass of the partners and $v$ the instantaneous velocity. The effective potential $U(r) + \mu v_0^2 b^2 / 2r^2$ in-
troduces a centrifugal energy barrier between the two collision partners (see Fig. 1). Since the kinetic energy $\mu v^2 / 2 \geq 0$, the
following condition must hold:

$$U(r) + \frac{\mu v_0^2 b^2}{2r^2} - \frac{1}{2}\mu v_0^2 \leq 0.  \tag{3}$$

The left hand side of Eq. (3) has its maximum at

$$r = \left(-\frac{\mu v_0^2 b^2 r_0^a}{Aa}\right)^{1/(2+a)}.  \tag{4}$$

If the incoming collision partner can cross this critical distance, where the centrifugal barrier has its maximum, a collision
leading to cluster formation will occur.

Inserting Eq. (4) into Eq. (3), we obtain the maximum impact parameter $b_{\max}$ for which a collision is still possible, which
can then be used to express the collision cross section $\Omega_{\mathrm{CF}}$ in an ideal, isotropic, central field as

$$\Omega_{\mathrm{CF}}(v_0) = \pi b_{\max}^2 = \frac{\pi a}{a+2}\left(-\frac{\mu v_0^2}{A(a+2)}\right)^{2/a} r_0^2.  \tag{5}$$





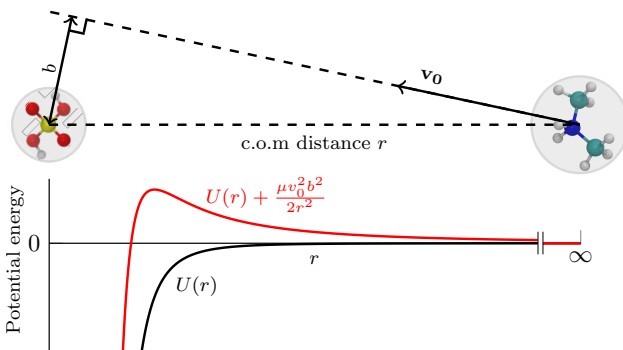

**Figure 1.** A schematic diagram of the central field approach with the corresponding potential energy profile. The molecule on the left is at rest and its center of mass (c.o.m.; center of the gray circle) designates the center of the field, while the molecule on the right initially moves along a trajectory set by velocity vector $\mathbf{v_0}$. At the start, the colliding molecules are infinitely far away from each other and the intermolecular potential energy $U(r)$ (black curve) equals zero. The perpendicular distance between the trajectory and the center of the field is the impact parameter $b$. If $b > 0$, the orbital angular momentum gives rise to a centrifugal barrier shown by the red curve.

In thermal equilibrium, the initial velocity $v_0$ follows the Maxwell-Boltzmann distribution, $f_{\mathrm{MB}}(v)$. For interaction exponent $a < -2$, the collision rate coefficient $\beta_{\mathrm{CF}}$ in a central field can be calculated as

$$
\begin{aligned}
\beta_{\mathrm{CF}}(T) &= \int_0^\infty \mathrm{d}v_0 \, v_0 f_{\mathrm{MB}}(v_0)\Omega_{\mathrm{CF}}(v_0) \\
&= \pi r_0^2 \sqrt{\frac{8k_{\mathrm{B}}T}{\pi\mu}} \Gamma\left(\frac{2+a}{a}\right)\left(-\frac{2k_{\mathrm{B}}T}{A(a+2)}\right)^{2/a},
\end{aligned}
\tag{6}
$$

where $k_{\mathrm{B}}$ is the Boltzmann constant, $T$ is the temperature, and $\Gamma(x)$ denotes the Gamma function of $x$.

The presented central field model is essentially *adiabatic*; it is assumed that there is no exchange of energy between rotational and vibrational modes of the collision partners, or exchange of angular momentum between the rotations of the collision partners and the orbiting motion of the system as a whole (Su and Bowers, 1973). For actual chemical compounds with internal structures, strong interactions can affect the rotational motion of the molecules which effectively changes the height of the centrifugal barrier as the angular momentum of the system is conserved.

The expressions for both the collision cross section and rate coefficient are derived for a general, well-behaving, asymptotic attractive interaction given by Eq. (1), and hence Eqs. (5) and (6) are convenient expressions to analyze and characterize the collision dynamics. In addition, two well-known results can be directly derived from Eq. (6): (1) When the interaction exponent $a$ approaches $-\infty$, the rate coefficient $\beta_{\mathrm{CF}}$ reduces to the kinetic gas theory result for two hard spheres of radii $R_i$ and $R_j$:

$$
\beta_{\mathrm{HS}}(T) = \pi(R_i + R_j)^2 \sqrt{\frac{8k_{\mathrm{B}}T}{\pi\mu}}.
\tag{7}
$$





(2) The main contribution to the intermolecular interactions for collisions between an ion and neutral particle is the ion-induced dipole interaction,

$$U(r) = \frac{\alpha q^2}{2r^4},$$  (8)

where $q$ is the charge of the ion and $\alpha$ is the angle-averaged polarizability of the dipole. For such an interaction, Eq. (6) becomes

$$\beta_\mathrm{L} = 2\pi q \sqrt{\frac{\alpha}{\mu}}.$$  (9)

This is known as the Langevin-Gioumousis-Stevenson expression (Langevin, 1905; Gioumousis and Stevenson, 1958). Note that the temperature dependency of $\beta_\mathrm{L}$ vanishes because the interaction exponent $a = -4$ in Eq. (8).

## 2.1 Su and Chesnavich parametrization

For collisions between an ion and polar neutral compound, angle-dependent ion-permanent dipole interactions should also be considered. In the most extreme case, the orientation of the dipole can be "locked in" so that the strength of the interaction is maximized. While thermal rotations of the collision partners will often prevent the dipole from "locking in", the ion-permanent dipole interaction is not necessarily averaged over all angles. Su and Chesnavich (1982) performed classical trajectory simulations of collisions between a point charge and a polarizable two-dimensional rigid rotor. Based on these findings, they developed a parametrized correction to the Langevin-Gioumousis-Stevenson expression (Eq. (9)):

$$\beta_\mathrm{SC} = K\beta_\mathrm{L}.$$  (10)

The correction term, $K$, depending only on the temperature, polarizability $\alpha$, and dipole moment $\mu_\mathrm{D}$, was found to be (Su and Chesnavich, 1982)

$$K = \begin{cases} 0.4767x + 0.6200; & x \geq 2, \\ \frac{(x+0.5090)^2}{10.526}; & x \leq 2 \end{cases}$$  (11)

with

$$x = \frac{\mu_\mathrm{D}}{(2\alpha k_\mathrm{B} T)^{1/2}}.$$  (12)

Su and Chesnavich observed that for all realistic systems, $K$ does not depend on the moments of inertia of the collision partners (Su and Chesnavich, 1982). Note that Eqs. (8)–(12) are written for Gaussian cgs units.

Maergoiz et al. (1996a) later validated Eq. (11) with their independent trajectory study of a similar system.

## 2.2 Atomistic model of ion-dipole systems

### 2.2.1 Collision systems

We studied a total of eight ion-dipole collision systems. Systems with only molecular ions and dipoles were studied, as well as systems with either a dipolar or charged dimer, or both. For five of the systems, sulfuric acid ($H_2SO_4$) served as the molec-



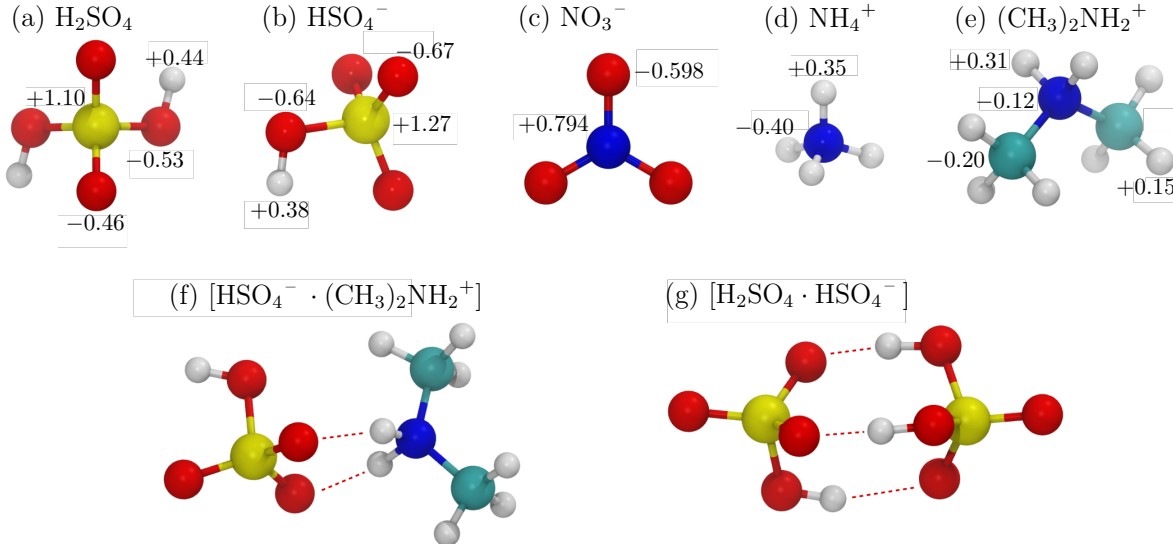

**Figure 2.** Stick-and-ball representations of the studied dipoles and ions: (a) sulfuric acid molecule, (b) bisulfate, (c) nitrate, (d) ammonium, (e) dimethylammonium ions, (f) neutral bisulfate-dimethylammonium dimer, and (g) charged sulfuric acid-bisulfate dimer. Atom partial charges according to the force field are indicated in panels a-e. The color codes of the atoms are as follows: sulfur–yellow, oxygen–red, nitrogen–blue, carbon–cyan, and hydrogen–white. Hydrogen bonds in the dimer structures (f,g) are indicated by dashed red lines.

ular dipole, while the ion was (1) bisulfate ($HSO_4^-$), (2) nitrate ($NO_3^-$), (3) ammonium ($NH_4^+$), (4) dimethylammonium ($(CH_3)_2NH_2^+$), and (5) the sulfuric acid-bisulfate dimer ($[H_2SO_4 \cdot HSO_4^-]$). The remaining three systems had the bisulfate-

dimethylammonium dimer ($[HSO_4^- \cdot (CH_3)_2NH_2^+]$) as a dipolar dimer. For these three systems, the ions were (1) $HSO_4^-$, (2) $(CH_3)_2NH_2^+$, and (3) $[H_2SO_4 \cdot HSO_4^-]$. The ions and dipoles of the studied systems are illustrated in Fig. 2 and their key physical properties are provided in Appendices A and B.

### 2.2.2   Force field

To describe the test systems, we employed a force field fitted according to the OPLS all-atom procedure (Jorgensen et al.,

1996). In the OPLS force field, the intramolecular interactions consist of harmonic bond potentials between covalently bonded atoms, harmonic angle potentials between atoms separated by two covalent bonds, and dihedral angle potentials between atoms separated by three covalent bonds,

$$U_{intra}^{OPLS} = \sum_{i=1}^{N_{bonds}} \frac{k_i^b}{2} \left( r_i - r_i^0 \right)^2 + \sum_{j=1}^{N_{angles}} \frac{k_j^\theta}{2} \left( \theta_j - \theta_j^0 \right)^2$$

$$+ \sum_{k=1}^{N_{dihedrals}} \sum_{n=1}^{4} \frac{V_n}{2} \left[ 1 + \cos \left( n\phi^k - \phi_n^k \right) \right], \tag{13}$$





where $k_i^{\mathrm{b}}$, $r_i$, and $r_i^0$ are the force constant, instantaneous, and equilibrium length of bond $i$, $k_j^\theta$, $\theta_j$, and $\theta_j^0$ are the force constant, instantaneous, and equilibrium value of angle $j$, and $V_n$, $\phi_n^k$, and $\phi^k$ are the Fourier coefficients, phase angles, and instantaneous value of the dihedral angle $k$.

The intermolecular interactions, as well as intramolecular interactions between atoms separated by more than three covalent bonds, are described by Lennard-Jones potentials between atoms $i$ and $j$ separated by a distance $r_{ij}$, with distance and energy

parameters $\sigma_{ij}$ and $\epsilon_{ij}$, and Coulomb interactions between the atoms' partial charges $q_i$ and $q_j$,

$$
\begin{aligned}
U_{\mathrm{inter}} = {} & \sum_{i=1}^{N_1}\sum_{j=1}^{N_2} 4\epsilon_{ij}\left[\left(\frac{\sigma_{ij}}{r_{ij}}\right)^{12} - \left(\frac{\sigma_{ij}}{r_{ij}}\right)^{6}\right] \\
& + \sum_{i=1}^{N_1}\sum_{j=1}^{N_2}\frac{1}{4\pi\epsilon_0}\frac{q_i q_j}{r_{ij}},
\end{aligned}
\tag{14}
$$

where $\epsilon_0$ is the vacuum permittivity.

The OPLS force field parameters used in this study were obtained from Loukonen et al. (2014) for $H_2SO_4$, $HSO_4^-$, and

$(CH_3)_2NH_2^+$ and from Mosallanejad et al. (2020) for $NO_3^-$ and $NH_4^+$. We note that in the original OPLS force field, Lennard-Jones and Coulomb interactions between atoms separated by three covalent bonds ("1-4 interactions") are scaled by a factor 0.5. Loukonen et al. set this scaling factor to zero when fitting the force field parameters. For consistency, we have also set these interactions to zero in our simulations. The optimized geometry of the studied ions and dipoles described by the OPLS force field showed only minimal differences compared to *ab initio* geometries, obtained at the $\omega$B97X-D/6-31++G** level of

theory, taken from Elm (2019). Using the OPLS force field, we obtained dipole moments $\mu_{\mathrm{D}}^{\mathrm{OPLS}} = 3.17$ and 13.20 Debye, and polarizabilities $\alpha^{\mathrm{OPLS}} = 6.57$ and 7.91 Å$^3$, for $H_2SO_4$ and $[HSO_4^- \cdot (CH_3)_2NH_2^+]$, respectively. The agreement of these values with *ab initio* calculations is very good for $H_2SO_4$ and reasonable for $[HSO_4^- \cdot (CH_3)_2NH_2^+]$. The details of the dipole moment and polarizability calculations, as well as a benchmark of cluster binding energies, from *ab initio* and using the OPLS force field, are provided in Appendices A and B.

## 2.3 Potential of Mean Force

Temperature-dependent long-range attractive interactions and binding free energies of the ion-dipole systems can be obtained from the potential of mean force (PMF) as a function of the distance $r$ between the ion and the dipole. The PMF differs from the Helmholtz free energy profile by a term $-k_{\mathrm{B}}T\ln r^2$, which accounts for the configurational entropy of the system. PMFs were calculated with the well-tempered metadynamics method (Barducci et al., 2008), where the energy surface of a

system is explored along one or more collective variables (CVs). In order to explore the CV space systematically, during a molecular dynamics (MD) run, Gaussian energy packets are deposited at certain time intervals to make often visited regions around the energy minima less favorable. Eventually, the sum of the Gaussian packages converges to the negative of the PMF. In well-tempered metadynamics simulations, the height of the Gaussian packages is decreased over time to ensure proper convergence.





We ran well-tempered metadynamics simulations using the LAMMPS code (Plimpton, 1995) together with the PLUMED plug-in (Tribello et al., 2014). For each system, the distance between the centers of mass of the collision partners was used as the collective variable. To confine the systems to the non-asymptotic region of the PMF, harmonic upper walls along the CV at 32 Å, or 50 Å, were used for systems containing only molecular ions and dipoles, or at least one dimer, respectively. No cut-off was used for the Lennard-Jones and Coulomb potentials over the allowed range of the CV. To ensure that the dimer structures

remained intact during the PMF calculation, appropriate harmonic upper walls were also applied to the center-of-mass distance between their constituents. To speed up the calculations, we used 40 random walkers dropping Gaussian energy packages every 0.5 ps. For all systems, the energy packages had an initial width of 0.1 Å and initial height of $k_\mathrm{B}T$ and the bias factor was 25, except for the two most weakly binding systems, $H_2SO_4 - NH_4^+$ and $H_2SO_4 - (CH_3)_2NH_2^+$, where the initial height was $0.5k_\mathrm{B}T$ and the bias factor 15. We employed a Velocity Verlet integrator with a time step of 1 fs for a total simulation time of

4 ns per walker. A stochastic velocity rescaling thermostat with a time constant of 0.1 ps was used to maintain a temperature of $T = 300$ K. For the $H_2SO_4 - HSO_4^-$ system, PMF calculations were also carried out at $T = 200$ and 400 K, otherwise using similar well-tempered metadynamics parameters.

### 2.4 Molecular Dynamics collision simulations

To obtain ion-dipole collision cross sections and rate coefficients from MD simulations, we determined the collision probability

over a relevant range of impact parameters and relative velocities. All collision simulations were carried out with the LAMMPS code (Plimpton, 1995). At the start of the simulation, the collision partners were placed 600 Å apart along the $x$-axis, well beyond the cut-off of the Lennard-Jones and Coulomb potentials of the OPLS force field at 280 Å. The collision partners were first separately equilibrated for 50 ps using a Langevin thermostat with a damping factor of 0.1 ps. During the equilibration, both the center-of-mass motion of each collision partner and the angular momentum of the total system were removed. A

thorough analysis of different thermostats revealed that, for the studied flexible compounds, the Langevin thermostat is best suited to ensure equipartition of rotational and vibrational energies. Details of these investigations will be published elsewhere (Halonen et al., 2022). After the equilibration, the distance between the now orientationally randomized collision partners was decreased to 200 Å along the $x$-axis, bringing them within range of the long-range intermolecular potentials for impact parameters $b \lesssim 190$ Å. Both collision partners were then given a velocity along the $x$-direction of $v_x = \pm v_0/2$ towards each

other, where $v_0$ is the initial relative velocity. For each system, the range of relative velocities started at $50\,\mathrm{ms}^{-1}$ and increased in steps of $50\,\mathrm{ms}^{-1}$. The highest relative velocity was determined so that at least 99% of the Maxwell-Boltzmann distribution was sampled. We sampled impact parameters starting from 0 Å up to the first impact parameter for which the collision probability at all sampled relative velocities was zero, in steps of 1 Å along the $z$-axis.

Collisions were determined based on the minimum center-of-mass distance between the collision partners during the tra-

jectory. All collisions were simulated in the NVE ensemble, with an initial thermal energy corresponding to a temperature of 300 K achieved during equilibration. In addition, we studied the temperature dependence of the collision probability for the $H_2SO_4 - HSO_4^-$ system. We employed a Velocity Verlet integrator with a time step of 1 fs. The duration of the simulation was dependent on the initial relative velocity. It was determined as the time it would take for two non-interacting particles to





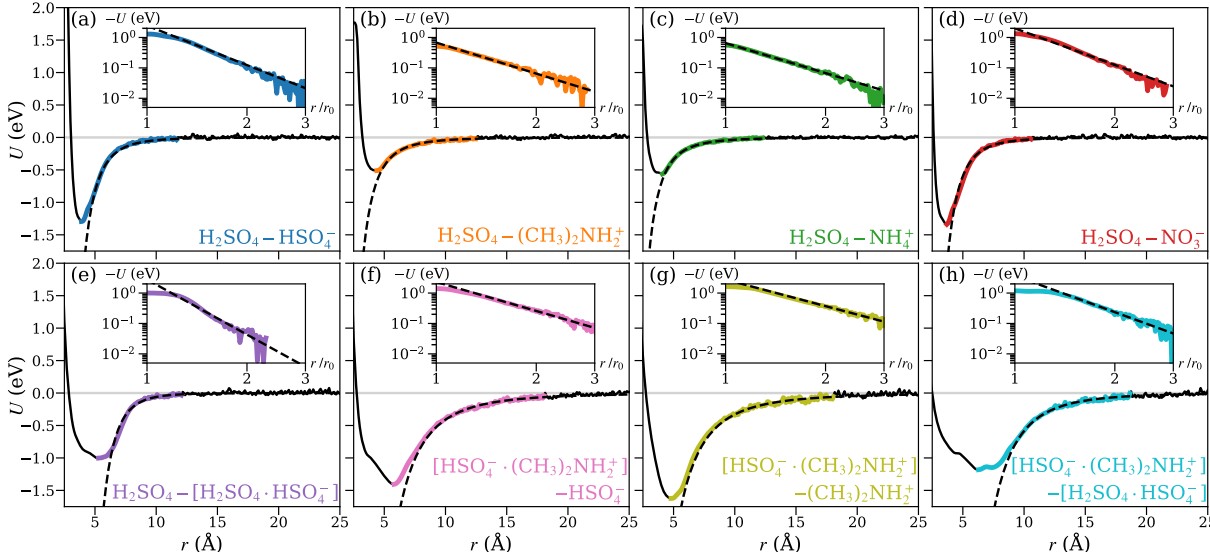

**Figure 3.** The potential of mean force (PMF) along the center-of-mass distance between the collision partners for the systems of $H_2SO_4$ and (a) $HSO_4^-$, (b) $(CH_3)_2NH_2^+$, (c) $NH_4^+$, (d) $NO_3^-$ (top row), and (e) $H_2SO_4$ and the $[H_2SO_4 \cdot HSO_4^-]$ dimer, the $[HSO_4^- \cdot (CH_3)_2NH_2^+]$ dimer and (f) $HSO_4^-$, (g) $(CH_3)_2NH_2^+$, (h) the $[H_2SO_4 \cdot HSO_4^-]$ dimer (bottom row). The solid black lines show the PMF curves obtained from well-tempered metadynamics simulations. The range over which the attractive interaction is fitted using Eq. (1) is highlighted in color, and shown on a logarithmic scale in the insets. The fits are shown as dashed black lines.

cross each other plus 50 ps. For each combination of relative velocity and impact parameter, 1000 independent trajectories
were simulated to obtain a statistically significant estimate of the collision probability.

## 3 Results and Discussion

### 3.1 Long-range attractive interactions fitted to the potential of mean force

Figure 3 shows the potentials of mean force (PMF) as a function of the center-of-mass distance between the collision partners, for the eight studied systems, obtained from well-tempered metadynamics simulations. We fitted the general intermolecular
interaction potential of Eq. (1) to each of the PMF curves. The distance parameter $r_0$ was set to the location of the minimum of the PMF curve. It should be noted that for the long-range attractive tail of the potential, this choice does not influence the final results when applying the central field model as $r_0$ only affects the coefficient $A$, while leaving the exponent $a$ unaltered. The attractive interaction was fitted to the range at which $U(r)$ is continuously negative and $r > r_0$, shown in the insets of Fig. 3 on logarithmic scale, so that the slope of the inset curve, $\ln(-U)$, is the interaction parameter $a$ and the $y$-intersect is $\ln A$. The
fitted values of the interaction parameters $r_0$, $A$, and $a$ are summarized in Tab. 1.



**Table 1.** The location of the potential minimum $r_0$, the interaction coefficient $A$, and the interaction exponent $a$, obtained by fitting Eq. (1) to the potential of mean force, as well as the reduced mass $\mu$ and the $x$ parameter in the Su and Chesnavich parametrization, for each studied ion-dipole system.

| System | $r_0$ (Å) | $A$ (eV) | $a$ | $\mu$ (amu) | $x$ |
|---|---|---|---|---|---|
| $H_2SO_4 - HSO_4^-$ | 3.89 | 2.45 | -4.31 | 48.75 | 4.30 |
| $H_2SO_4 - NO_3^-$ | 3.69 | 2.00 | -4.00 | 37.98 | 4.30 |
| $H_2SO_4 - (CH_3)_2NH_2^+$ | 4.39 | 0.68 | -3.38 | 31.31 | 4.30 |
| $H_2SO_4 - NH_4^+$ | 4.09 | 0.65 | -3.27 | 15.21 | 4.30 |
| $H_2SO_4 - [H_2SO_4 \cdot HSO_4^-]$ | 5.29 | 2.78 | -6.01 | 65.22 | 4.30 |
| $[HSO_4^- \cdot (CH_3)_2NH_2^+] - HSO_4^-$ | 5.79 | 2.31 | -3.16 | 57.80 | 18.52 |
| $[HSO_4^- \cdot (CH_3)_2NH_2^+] - (CH_3)_2NH_2^+$ | 4.79 | 2.77 | -2.90 | 34.80 | 18.52 |
| $[HSO_4^- \cdot (CH_3)_2NH_2^+] - [H_2SO_4 \cdot HSO_4^-]$ | 6.28 | 3.71 | -3.98 | 82.50 | 18.52 |

The fitted attractive interactions reveal interesting differences between the systems. The interactions between sulfuric acid ($H_2SO_4$) and the two anions are much stronger than with the two cations, in agreement with preliminary *ab initio* calculations (see Tab. B1). In addition, the exponent of the fitted interaction is close to $-4$ for the anions, similar to rotationally averaged ion-(induced) dipole interactions, but closer to $-3$ for the cations. The systems involving at least one dimer all exhibit strong

attractive interactions with interaction exponents between $-3$ and $-6$, indicating that those interactions are more complex, with the latter value resembling the standard interaction potential between permanent or induced dipoles. We note that for most systems considered here, the fitted effective interactions cannot be described by the standard ion-induced dipole, or ion-permanent dipole potential alone, but are a linear combination of different types of atomistic pair potentials.

### 3.2 Collision probability distributions and cross sections

Figures 4(a–d) and 5(a–d) show heat maps of the collision probabilities $P(v, b)$ obtained from the molecular dynamics (MD) collision simulations for molecular ion-dipole systems, and systems containing at least one dimer, respectively. The center-of-mass distance criterion for a successful collision was determined for each system by taking the distance at which the value of the PMF was $5k_BT$ ($\sim 0.13$ eV at 300 K) higher than its minimum, to account for thermal fluctuations. We tested multiple criteria for determining collisions, and found them to be quite robust. However, the chosen criterion was deemed the most

physically intuitive, as we are not interested in distinguishing between collisions and "sticking" in this work.

    Unlike in the PMF calculations, no constraints were imposed on center-of-mass distances within the dimer structures $[H_2SO_4 \cdot HSO_4^-]$ and $[HSO_4^- \cdot (CH_3)_2NH_2^+]$ in the MD collision simulations. In every single MD simulation, these dimers remained intact during equilibration. Evaporations of the original dimers after the collision were only observed for the $[HSO_4^- \cdot (CH_3)_2NH_2^+]$ dimer in a certain window of impact parameters at high relative velocities. By far the highest evaporation probability observed

was $\sim 3$ % for the $[HSO_4^- \cdot (CH_3)_2NH_2^+] - HSO_4^-$ system at $v = 1000$ ms$^{-1}$ and $b = 16$ Å. All collision trajectories resulting




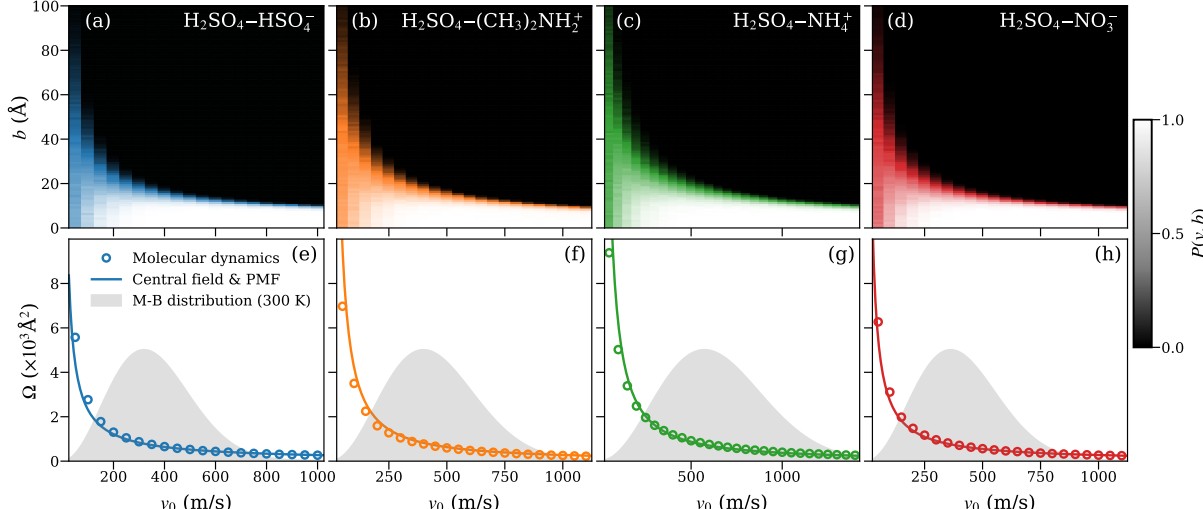

**Figure 4.** Heat maps of the collision probabilities from molecular dynamics (MD) for molecular ion-dipole systems as a function of initial relative velocity $v_0$ and impact parameter $b$ (a–d, top row). Corresponding dynamic collision cross sections $\Omega_{MD}(v)$ obtained from these MD collision probabilities (open circles) and collision cross sections $\Omega_{CF}(v)$ based on the central field model (solid lines) using an attractive interaction fitted to the PMF (e–h, bottom row). The Maxwell-Boltzmann distribution of the relative velocity for each system is indicated by the gray area.

in an evaporation of the original dimer were discarded and additional simulations carried out to ensure a valid sample size of 1000 for each combination of relative velocity and impact parameter.

The collision probability heat maps all exhibit similar dependencies on $v$ and $b$: for large relative velocities, the collision probability is unity at small impact parameters $b \lesssim 10$ Å, and drops sharply to zero when $b$ increases. As the relative velocity

decreases, the decline in collision probability is shifted to larger values of $b$, while at the same time becoming more gradual as a function of $b$. At small relative velocities ($v_0 \lesssim 200$ ms$^{-1}$), we observe collision probabilities significantly lower than one for small values of $b$, while small, but non-zero collision probabilities persist up to large values of $b$. This effect is especially significant at the lowest values of $v_0$ considered, where the collision probability at $b = 0$ Å, is typically only $\sim 0.5$, but non-zero collision probabilities remain even past $b = 100$ Å. The reduced collision probabilities at low values of $v_0$ is caused by small

oscillations in the interaction energy, at large separation distances and periodic in time, which can lead to repulsion between the collision partners, as previously shown for dipole-dipole collisions (Halonen et al., 2019).

The dynamic collision cross section is a measure for the velocity dependent collision probability over all impact parameters considered in the MD simulations, and can be calculated as

$$\Omega_{MD}(v) = \pi \int_0^\infty db^2 \, P(v,b). \tag{15}$$



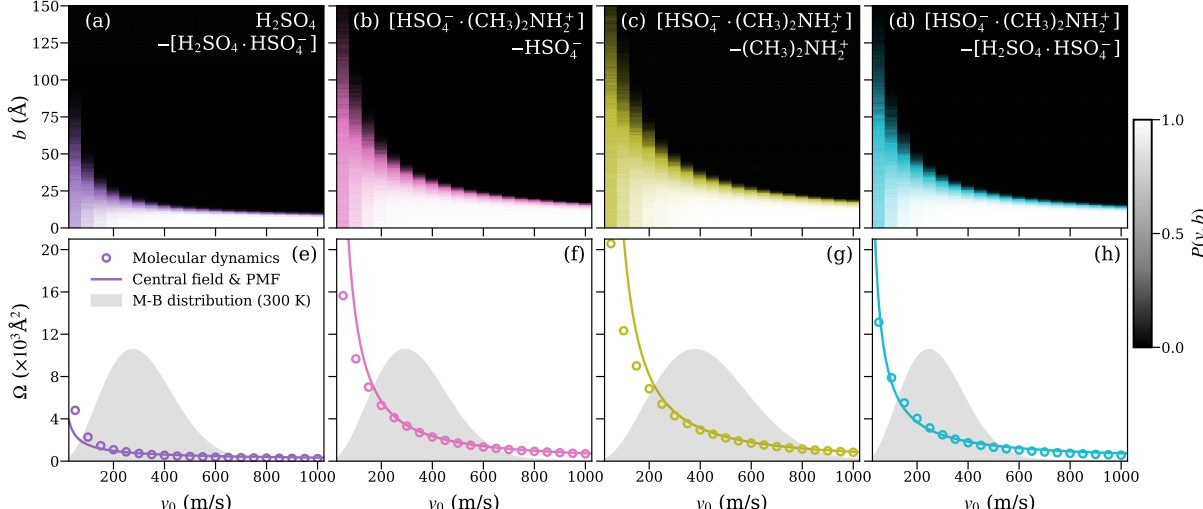

**Figure 5.** Heat maps of the collision probabilities from molecular dynamics (MD) for ion-dipole systems including at least one dimer as a function of initial relative velocity $v_0$ and impact parameter $b$ (a–d, top row). Corresponding dynamic collision cross sections $\Omega_{MD}(v)$ obtained from these MD collision probabilities (open circles) and collision cross sections $\Omega_{CF}(v)$ based on the central field model (solid lines) using an attractive interaction fitted to the PMF (e–h, bottom row). The Maxwell-Boltzmann distribution of the relative velocity for each system is indicated by the gray area.

Figures 4(e–h) and 5(e–h) show the collision cross section obtained from the MD collision simulations of molecular systems, and systems containing at least one dimer, respectively, as circles. The collision cross section from the central field model, evaluated using Eq. (5) with parameters $r_0$, $A$, and $a$, obtained from the fit to the PMF curve of the system, is given by the solid line. For the molecular ion-dipole systems, we find excellent agreement between collision cross sections obtained from the collision MD simulations and from the central field model using the fit to the PMF, over the entire range of relative velocities.

For the systems containing at least one dimer, the agreement between the two approaches is still good, in particular over the most relevant velocity range according to the Maxwell-Boltzmann distribution.

  This agreement indicates that the framework of the central field model adequately captures the underlying atomistic collision dynamics of the studied ion-dipole systems. As the model assumes point-like, structureless collision partners, it disregards any energy transfer between the partners' respective internal degrees of freedom and the orbiting motion of the system as a

whole. As a result, the centrifugal barrier is solely determined by the system's initial orbital angular momentum. Based on the demonstrated predictive power of Eq. (5), the assumption of adiabaticity (i.e., that the motion of the orbiting system as a whole is well-separated from the intramolecular motion of the collision partners) is well justified here. Furthermore, the results imply that the effective field of attraction is isotropic and can be described by simple interaction parameters, even for collision partners with complex and dynamic charge distributions, such as the $[HSO_4^- \cdot (CH_3)_2NH_2^+] - [H_2SO_4 \cdot HSO_4^-]$ system.



### 3.3 Canonical collision rate coefficients

The canonical collision rate coefficient can be calculated from the effective collision cross section of Eq. (15) as

$$\beta_{\mathrm{MD}} = \pi \int \mathrm{d}v \, v f_{\mathrm{MB}}(v) \Omega_{\mathrm{MD}}(v). \tag{16}$$

Table 2 shows the collision rate coefficients, obtained from the MD collision simulations $\beta_{\mathrm{MD}}$ in Eq. (16), the central field model with interaction parameters fitted to the PMF curve $\beta_{\mathrm{CF}}$ in Eq. (6), the Langevin-Gioumousis-Stevenson model $\beta_{\mathrm{L}}$ in Eq. (9), and the Su and Chesnavich parametrization $\beta_{\mathrm{SC}}$ in Eq. (11), for all studied systems. Overall, we find collision rate coefficients of similar magnitudes for the systems containing a molecular dipole. In comparison, systems containing a dipolar dimer exhibit larger collision rate coefficients as well as more variation in magnitude between the systems. We find excellent agreement within 10 % across all systems for the collision rate coefficients obtained from MD and the central field model using the interactions obtained from the PMF calculation. This indicates that collisions dynamics are indeed well captured by an adiabatic model and isotropic interactions.

The Langevin-Gioumousis-Stevenson model performs quite poorly with collision rate coefficients deviating by a factor of three for systems with a molecular dipole and a factor larger than 12 for systems containing a dipolar dimer. In comparison, the Su and Chesnavich parametrization is in quite good agreement with the MD or central field results, underestimating the collision rate coefficients by 10–20 % for systems containing $H_2SO_4$ as the dipole, and 30–40 % for systems containing $[HSO_4^- \cdot (CH_3)_2NH_2^+]$. The Su and Chesnavich parametrization differs from the Langevin-Gioumousis-Stevenson expression in two ways: first, in addition to the polarizability it also takes into account the permanent dipole moment of the neutral collision partner. Second, the thermal correction factor, obtained by Su and Chesnavich from trajectory simulations, adds a dependence on temperature to account for the dynamics of the rotating dipole. This significantly improves the accuracy of the model compared to the Langevin-Gioumousis-Stevenson expression and explains the rather good agreement with the all-atom molecular dynamics simulations in this work. However, we expect the discrepancies to increase for more complex systems.

Due to the Maxwell-Boltzmann distribution of velocities, the collision rate coefficients given by the central field model and the Su and Chesnavich parametrization are proportional to $\mu^{-1/2}$. In Fig. 6, we show the collision rate coefficients as a function of $\mu^{-1/2}$ for all systems, from molecular dynamics trajectory simulations (MD), the central field model using the attractive potential fitted to the PMF (CF), and the Su and Chesnavich parametrization (SC). Strikingly, for systems where $H_2SO_4$ is the neutral collision partner, the collision rate coefficients are almost exactly proportional to $\mu^{-1/2}$. For the three collisions involving $[HSO_4^- \cdot (CH_3)_2NH_2^+]$, where the differences between $\beta_{\mathrm{MD}}$ and $\beta_{\mathrm{CF}}$, and $\beta_{\mathrm{SC}}$ are more pronounced, a somewhat linear trend with $\mu^{-1/2}$ is still observed. Thus, the main property of the ion affecting $\beta$ is its mass. This simple correlation between $\beta$ and $\mu$ is rather unexpected due to the notable difference in the interaction potential (presented in Fig. 3). For the central field model, the results suggest that the effect of different interaction parameters ($A$, $a$ and $r_0$) is balanced out in Eqs. (5) and (6), which is reflected in the very similar collision cross sections for the different ion–$H_2SO_4$ systems presented in Fig. 4(e–h), even though their underlying interaction parameters differ from each other, as shown in Tab. 1. Further research is needed to determine the underlying reason for this invariance.





**Table 2.** Collision rate coefficients obtained from molecular dynamics collision simulations $\beta_{\mathrm{MD}}$, central field model with interaction parameters fitted to the PMF curve $\beta_{\mathrm{CF}}$, Langevin-Gioumousis-Stevenson model $\beta_{\mathrm{L}}$, and Su and Chesnavich (1982) parametrization $\beta_{\mathrm{SC}}$, for all studied systems in $10^{-15}$ m$^3$s$^{-1}$.

| System | $\beta_{\mathrm{MD}}$ | $\beta_{\mathrm{CF}}$ | $\beta_{\mathrm{MD}}/\beta_{\mathrm{CF}}$ | $\beta_{\mathrm{L}}$ | $\beta_{\mathrm{MD}}/\beta_{\mathrm{L}}$ | $\beta_{\mathrm{SC}}$ | $\beta_{\mathrm{MD}}/\beta_{\mathrm{SC}}$ |
|---|---|---|---|---|---|---|---|
| $H_2SO_4 - HSO_4^-$ | 2.62 | 2.51 | 1.04 | 0.81 | 3.21 | 2.17 | 1.20 |
| $H_2SO_4 - (CH_3)_2NH_2^+$ | 3.01 | 3.28 | 0.92 | 1.02 | 2.96 | 2.71 | 1.11 |
| $H_2SO_4 - NH_4^+$ | 4.43 | 4.23 | 1.05 | 1.46 | 3.04 | 3.89 | 1.14 |
| $H_2SO_4 - NO_3^-$ | 2.85 | 2.73 | 1.04 | 0.92 | 3.09 | 2.46 | 1.16 |
| $H_2SO_4 - [H_2SO_4 \cdot HSO_4^-]$ | 2.18 | 2.21 | 0.98 | 0.70 | 3.10 | 1.88 | 1.16 |
| $[HSO_4^- \cdot (CH_3)_2NH_2^+] - HSO_4^-$ | 9.50 | 10.25 | 0.93 | 0.72 | 13.15 | 6.83 | 1.39 |
| $[HSO_4^- \cdot (CH_3)_2NH_2^+] - (CH_3)_2NH_2^+$ | 11.62 | 12.91 | 0.90 | 0.93 | 12.48 | 8.80 | 1.32 |
| $[HSO_4^- \cdot (CH_3)_2NH_2^+] - [H_2SO_4 \cdot HSO_4^-]$ | 7.50 | 7.40 | 1.01 | 0.60 | 12.40 | 5.72 | 1.31 |

## 3.4 Temperature dependence

To study the temperature dependence of the collision kinetics, and the extent to which this is captured by the theoretical
models, we performed additional MD trajectory simulations and PMF calculations, for the $H_2SO_4$–$HSO_4^-$ system at $T = 200$
and 400 K. Figure 7 shows the PMFs at different temperatures, along with the fits of the attractive interactions for the central
field model at the given temperature. Over the temperature range considered, the position of the energy minimum ($r_0$) remains
unaffected, which implies that the binding mechanism remains the same in this temperature range. The depth of the potential
well, on the other hand, decreases with increasing temperature, and the changes in the shape of the potentials are reflected in

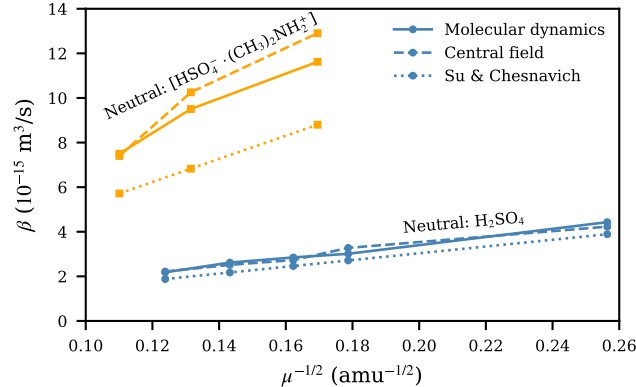

**Figure 6.** Collision rate coefficient $\beta$ plotted as a function of $1/\sqrt{\mu}$ for collisions involving $H_2SO_4$ or $[HSO_4^- \cdot (CH_3)_2NH_2^+]$, from molecular dynamics trajectory simulations, the central field model using the attractive potential fitted to the PMF, and the Su and Chesnavich (1982) parametrization.





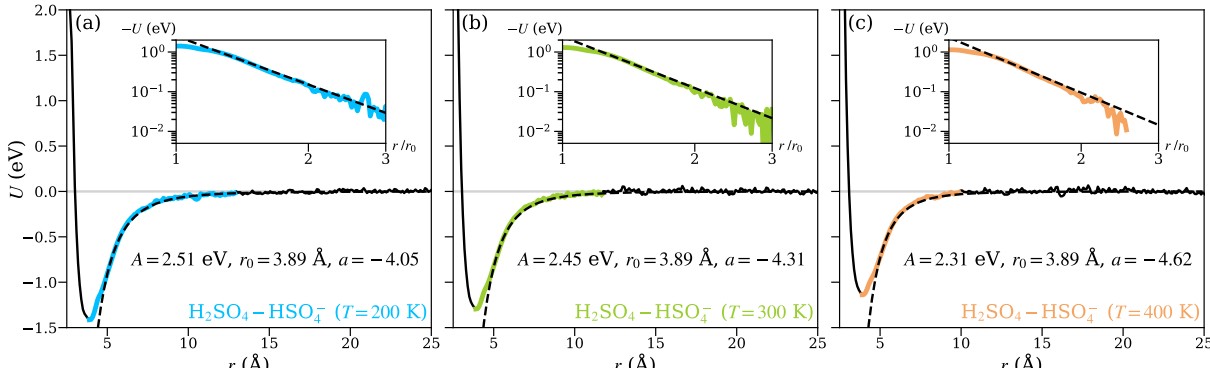

**Figure 7.** The potential of mean force (PMF) along the center-of-mass distance between the collision partners for the $H_2SO_4$–$HSO_4^-$ system at temperatures (a) 200 K, (b) 300 K, and (c) 400 K. The solid black lines show the PMF curves obtained from well-tempered metadynamics simulations. The range over which the attractive interaction is fitted using Eq. (1) is highlighted in color, and shown on a logarithmic scale in the insets. The fits are shown as dashed black lines.

a small, but significant, decrease of the values of the parameters $A$ and $a$. The almost linear dependence of the well depth on temperature is mostly due to increasing thermal motion in the molecules' rotational and vibrational degrees of freedom.

     Figure 8(a) shows the collision cross sections obtained from MD simulations using different equilibration temperatures, sampling the relevant $(v_0, b)$-space (at $T = 400$ K, the relative velocity range was extended from 1000 to 1100 ms$^{-1}$ to cover a wider velocity distribution). The central field model, using the fits to the PMFs at different temperatures, agrees with the

collision cross sections from MD simulations, and presents a similar narrowing trend of the cross sections with increasing temperature. The agreement is especially good at $T = 200$ K, but slightly worsens as temperature increases. The temperature dependence of the collision cross sections shows that the thermal energy of the colliding partners significantly affects the actual collision trajectories, and this effect is well captured by the changes in the potentials of mean force at different temperatures.

     Due to the temperature dependence of the PMFs, correct collision rate coefficients cannot be computed by simply temperature-

scaling $\beta_{CF}$ according to the Maxwell-Boltzmann distribution while using fixed parameters $A$ and $a$ obtained at a certain reference temperature. This is shown in Fig. 8(b), where the temperature dependencies of scaled $\beta_{CF}$ (solid lines), based on the three PMF calculations at different temperatures are shown alongside the $\beta_{MD}$ results (markers) at these temperatures. While Eq. (6) predicts a small positive temperature dependence for $a \lesssim -4$, the collision rate coefficients from MD simulations show that the actual dependence is stronger and negative. Scaling collision rate coefficients obtained from PMFs at a specific temper-

ature to other temperatures thus leads to significant discrepancies. In contrast, when using parameters from PMF calculations performed at the correct temperature, the central field model does predict a similar trend for collision rate coefficients as the MD simulations, as shown in Fig. 8(b). The agreement worsens at higher temperatures, with $\beta_{MD}/\beta_{CF}$ becoming about 1.14 at $T = 400$ K . Finally, while the Su and Chesnavich parametrization systematically underestimates the collision rate coefficients, the temperature dependence shows a similar trend as in the MD simulations over the range of temperatures considered. This




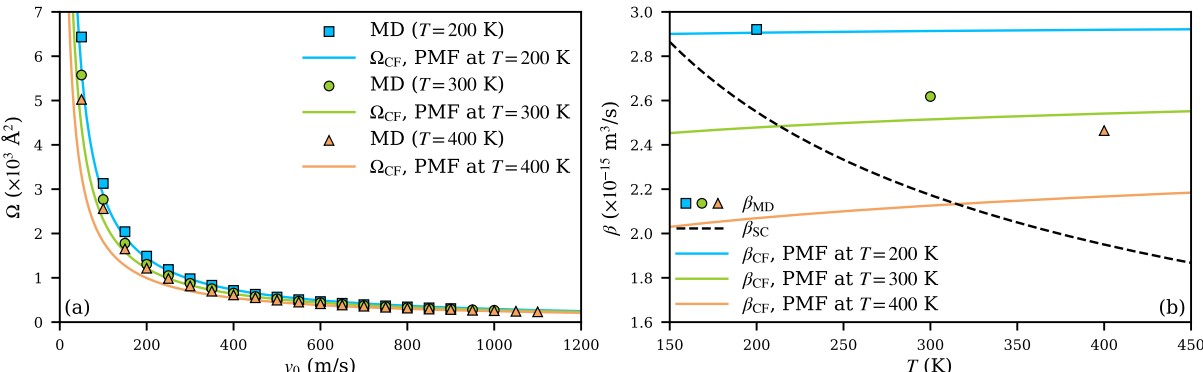

**Figure 8.** The collision cross section $\Omega$ (a) and collision rate coefficient $\beta$ (b) of the $H_2SO_4$-$HSO_4^-$ system at temperatures 200 K (blue), 300 K (green), and 400 K (orange). Results are shown as squares, circles, and triangles at the respective temperatures for MD simulations, as solid lines for the central field model, and as a dashed line for the Su and Chesnavich (1982) model.

good qualitative performance is due to the fact that the parametrization is based on trajectory calculations and thus is able to describe the fundamental dynamical effects adequately, if not the detailed interactions between actual molecules.

## 4 Conclusions

A proper theoretical treatment of bimolecular reactions requires an accurate assessment of the intermolecular potential. In the context of atmospheric clusters and their formation, high-level *ab initio* calculations are necessary for assessing the clusters'
stability in equilibrium (Elm et al., 2020). However, collision processes are governed by a centrifugal barrier located at larger intermolecular distances, where the strength of the interaction between colliding partners can be described with satisfactory accuracy and very small computational cost using classical force fields. In this study we have demonstrated the ability of different modelling approaches to describe the collision dynamics of ion-dipole systems in the free molecular regime.

The demand for accurate theoretical modelling of collisions between atmospheric molecules and clusters (neutral or charged)
arises from several recent observations and considerations: (1) At polluted sites, new particle formation (NPF) is controlled predominantly by collisions due to high vapor concentrations and extremely stable dimers (Kürten et al., 2014; Xiao et al., 2021). (2) Due to the immense improvement in *ab initio* calculations (Elm et al., 2020; Elm, 2020), the accuracy of current kinetic models used to predict atmospheric NPF rates is increasingly limited by the estimates for the collision rates, rather than the evaporation rates (Jiang et al., 2022). (3) In chemical ionization mass spectrometry, collisions between a studied
atmospheric cluster and a charging ion lead to the formation of a detectable charged cluster in the ionization chamber, while non-sticking collisions between the charged cluster and residual carrier gas in the atmospheric pressure interface can lead to cluster fragmentation causing systematic errors in the mass spectra (Zapadinsky et al., 2018). (4) One reason that ion-induced NPF is sometimes disregarded in global aerosol models is the lack of accurate rate coefficients for charged clusters (He et al., 2020).





While simple analytical and parametrized models exist for the calculation of collision rate coefficients of ion-dipole systems, these models do not directly account for the complexity encountered in molecular ions and dipoles, let alone dipolar or charged clusters. Thus, in this study we considered two fundamentally different modelling approaches to calculate ion-dipole collision rates of atmospheric molecules and clusters in the free molecular regime: molecular dynamics trajectory simulations and a central field model. Since accurate experimental data is currently missing for the investigated systems, the presented MD

simulations serve as reference as each collision trajectory evolves under the full set of atomistic interactions defined by the force field. In contrast, the motion of colliding particles in the central field model follows from the assumption that the effective interaction potential is close to isotropic and adiabatic, and thus provides an adequate analytical solution for a certain group of systems at specific conditions.

In this work, we studied collisions of eight atmospherically relevant ion-dipole systems, described by an atomistic OPLS-

based force field. To achieve comparability between the MD simulations and the central field modelling, the attractive interaction in the central field model was fitted to the potentials of mean force between the collision partners, obtained from well-tempered metadynamics calculations at the respective temperature. The velocity-dependent collision cross sections from the central field model and the molecular dynamics simulations were found to be in excellent agreement, supporting the assumption that the process can be described by isotropic and adiabatic intermolecular dynamics. Furthermore, the cross sections

are very similar for systems with the same neutral dipole despite the differences in the underlying interaction potentials. Thus, we concluded that for the studied systems the canonical collision rate coefficients depend mostly on the dipole's properties while the ion affects only the velocity distribution, through its mass. This finding and the collision rate coefficients calculated from atomistic simulations are in good agreement with the widely used parametrization by Su and Chesnavich (1982) which is based on simplified trajectory simulations of point charges and polar rods and the classical (temperature-independent)

Langevin-Gioumousis-Stevenson (Langevin, 1905; Gioumousis and Stevenson, 1958) model. The dynamical correction given by the parametrization significantly improves the prediction of the Langevin-Gioumousis-Stevenson model, and the temperature dependence of the parametrization is found to be in qualitative agreement with the MD simulations. However, we found that the Su and Chesnavich (1982) parametrization predicts the collision rate coefficients less accurately for systems with a dimer as the dipolar collision partner. This inaccuracy may become even more pronounced for systems involving larger clusters,

which need to be investigated in future research.

We have demonstrated that the combination of PMF calculation and central field model is a viable and elegant alternative to brute force sampling of collision trajectories over a large range of impact parameters and relative velocities, in particular for systems with long-ranged attractive interactions, such as between ions and dipoles in the gas-phase. The presented approaches will be used in the future to obtain the collision rate coefficients of a large group of molecules and clusters. The resulting

data will allow the assessment of the relative importance of particle growth pathways involving ions in the initial stages of atmospheric new particle formation (He et al., 2020).



**Appendix A: *Ab initio* and force field calculations of dipole moments, and average polarizabilities**

*Ab initio* values for the dipole moment and polarizability of $H_2SO_4$ and $[HSO_4^- \cdot (CH_3)_2NH_2^+]$ presented in Tab. A1 were obtained with the Gaussian 16 program (Frisch et al., 2016). Geometry optimizations were done with the $\omega$B97X-D functional

(Chai and Head-Gordon, 2008) and the 6–31++G(d,p) basis set. The starting geometries for $H_2SO_4$ and $[HSO_4^- \cdot (CH_3)_2NH_2^+]$ were taken from the Atmospheric Cluster Database of Elm (2019).

For the force field model, the dipole moments $\mu_D$ of the collision partners studied in this work, reported in Tab. A1, were calculated for energy minimized configurations as

$$\mu_D = \left\| \sum_{i=1}^{N_c} (\mathbf{r}_i - \mathbf{r}_{com}) q_i \right\|, \tag{A1}$$

where $N_c$ is the number of partial charges $q_i$ with positions $\mathbf{r}_i$ in the compound, and $\mathbf{r}_{com}$ is the position of the compound's center of mass.

To determine the average polarizability of the compounds $\bar{\alpha}$, reported in Tab. A1, we applied electric fields $E_j$ of different signs and magnitudes along the three Cartesian axes and performed an energy minimization of the system in LAMMPS (Plimpton, 1995), and recorded the resulting changes in the dipole moment components, $\Delta \mu_{D,i}$. We then performed linear fits

around $\|E_j\| = 0$ to obtain the components of the polarizability tensor as

$$\alpha_{ij} = \frac{\Delta \mu_{D,i}}{E_j}. \tag{A2}$$

After diagonalizing the polarizability tensor, the average polarizability was calculated from the diagonal elements as

$$\bar{\alpha} = \frac{1}{3}(\alpha_{11} + \alpha_{22} + \alpha_{33}). \tag{A3}$$

**Appendix B: *Ab initio* and force field calculations of cluster binding energies**

We benchmarked the accuracy of the all-atom OPLS force field in describing the structures and energies of stable clusters formed upon ion-dipole collision against *ab initio* calculations.

*Ab initio* single-point energies of all collision partners, and clusters formed after collision, were obtained from the Atmospheric Cluster Database of Elm (2019), except for $NO_3^-$, $H_2SO_4 - NO_3^-$, and $[HSO_4^- \cdot (CH_3)_2NH_2^+] - (CH_3)_2NH_2^+$, as these were not available from the database. The geometries of the compounds in the database were optimized using the $\omega$B97X-D

functional (Chai and Head-Gordon, 2008) and the 6–31++G(d,p) basis set in the Gaussian 09, rev. D.01 program (Frisch et al., 2013). A single point energy calculation using the DLPNO-CCSD(T) (Riplinger and Neese, 2013; Riplinger et al., 2013) with an aug-cc-pVTZ basis set and normal PNO settings (Liakos et al., 2015) was then performed on this optimized geometry.

For consistency with the single point energy values obtained from the database, we followed a similar procedure for $NO_3^-$, $H_2SO_4 - NO_3^-$, and $[HSO_4^- \cdot (CH_3)_2NH_2^+] - (CH_3)_2NH_2^+$. First, to identify the lowest free energy conformer of the $H_2SO_4 -$

$NO_3^-$ and $[HSO_4^- \cdot (CH_3)_2NH_2^+] - (CH_3)_2NH_2^+$ clusters, we used the JKCS configurational sampling procedure outlined by





**Table A1.** Dipole moments $\mu_{\mathrm{D}}^{\mathrm{OPLS}}$ and average polarizabilities $\bar{\alpha}^{\mathrm{OPLS}}$ of the chemical compounds used in the ion-dipole collision studies, as described by the all-atom OPLS force field. For charged compounds, the dipole moment is reported with respect to the compound's center of mass. For the neutral compounds, dipole moments $\mu_{\mathrm{D}}^{\mathrm{QM}}$ and average polarizabilities $\bar{\alpha}^{\mathrm{QM}}$ from *ab inito* calculation are provided for comparison.

| Compound | $\mu_{\mathrm{D}}^{\mathrm{OPLS}}$ (Debye) | $\mu_{\mathrm{D}}^{\mathrm{QM}}$ (Debye) | $\bar{\alpha}^{\mathrm{OPLS}}$ (Å$^3$) | $\bar{\alpha}^{\mathrm{QM}}$ (Å$^3$) |
|---|---|---|---|---|
| $H_2SO_4$ | 3.17 | 3.27 | 6.57 | 4.91 |
| $[HSO_4^- \cdot (CH_3)_2NH_2^+]$ | 13.20 | 8.97 | 7.91 | 10.13 |
| $HSO_4^-$ | 4.41 | | | |
| $NO_3^-$ | 0.00 | | | |
| $NH_4^+$ | 0.00 | | | |
| $(CH_3)_2NH_2^+$ | 1.84 | | | |
| $[H_2SO_4 \cdot HSO_4^-]$ | 4.92 | | | |

Conversion between cgs and SI units of polarizability:
$1\ \text{Å}^3 = 1.1126 \times 10^{-40}\ \text{Cm}^2\text{V}^{-1}$.

Kubečka et al. (2019). In this procedure, all possible conformers and conjugate acids and bases of the molecules in the cluster are included as rigid monomers. Every combination of these rigid monomers that satisfies the cluster composition and total charge is then determined. Using the genetic algorithm of the ABCluster program (Zhang and Dolg, 2015), we created 300 conformers of each of these combination by starting from a population of 4000, performing 100 genetic steps, and keeping

the 300 conformers lowest in energy. The resulting conformers were optimized with the GFN–$x$TB semi-empirical method (Grimme et al., 2017). After this optimization, duplicates were removed and all remaining conformers with electronic energy less than 20 kcal/mol above the lowest energy conformer at the GFN–$x$TB level of theory were further optimized, along with a calculation of their vibrational frequencies, using the $\omega$B97X-D functional and the 6–31++G(d,p) basis set in the Gaussian 16 program (Frisch et al., 2016). Of the three lowest free energy cluster conformers at the $\omega$B97X-D level of theory, we performed

a single point energy calculation, employing DLPNO-CCSD(T) with an aug-cc-pVTZ basis set and normal PNO settings, as was done for the Elm database. To calculate the binding energies, the DLPNO-CCSD(T) single-point energy of the conformer with the lowest Gibbs free energy, calculated as $G = E^{\mathrm{DLPNO}} + G_{\mathrm{corr}}^{\omega\mathrm{B97X-D}}$, was used.

     We used the global minimum energy configurations obtained from the *ab initio* calculations and performed an energy minimization with the all-atom OPLS force field in LAMMPS. For all compounds and clusters, the differences between the

re-optimized geometry and the *ab initio* reference structure were quite small, even for the larger clusters.

     The cluster binding energies from *ab initio* and using the OPLS force field reported in Tab. B1 are given as the difference between the energy of the formed cluster and the energies of the original collision partners. The energies are overall in quite good agreement, with the force field typically predicting slightly weaker binding energies. The average unsigned error for all eight systems is 0.21 eV, and the largest difference of +0.55 eV was observed for the $[HSO_4^- \cdot (CH_3)_2NH_2^+] - HSO_4^-$



**Table B1.** Binding energies of collision systems obtained from *ab initio* calculations, $\Delta E_{\mathrm{QM}}$ and using the OPLS force field, $\Delta E_{\mathrm{OPLS}}$, as well as the well depth of the potential of mean at $T = 300$ K, $\Delta U_{\mathrm{PMF}}$.

| System | $\Delta E_{\mathrm{QM}}$ (eV) | $\Delta E_{\mathrm{OPLS}}$ (eV) | $\Delta U_{\mathrm{PMF}}$ (eV) | $(\Delta E_{\mathrm{OPLS}} - \Delta E_{\mathrm{QM}})$ (eV) |
|---|---|---|---|---|
| $H_2SO_4 - HSO_4^-$ | $-2.04$ | $-1.70$ | $-1.29$ | $+0.34$ |
| $H_2SO_4 - (CH_3)_2NH_2^+$ | $-0.84$ | $-0.72$ | $-0.51$ | $+0.12$ |
| $H_2SO_4 - NH_4^+$ | $-0.92$ | $-0.77$ | $-0.56$ | $+0.15$ |
| $H_2SO_4 - NO_3^-$ | $-2.01$ | $-1.77$ | $-1.34$ | $+0.24$ |
| $H_2SO_4 - [H_2SO_4 \cdot HSO_4^-]$ | $-1.28$ | $-1.42$ | $-1.00$ | $-0.14$ |
| $[HSO_4^- \cdot (CH_3)_2NH_2^+] - HSO_4^-$ | $-1.93$ | $-1.38$ | $-1.41$ | $+0.55$ |
| $[HSO_4^- \cdot (CH_3)_2NH_2^+] - (CH_3)_2NH_2^+$ | $-1.83$ | $-1.80$ | $-1.62$ | $+0.03$ |
| $[HSO_4^- \cdot (CH_3)_2NH_2^+] - [H_2SO_4 \cdot HSO_4^-]$ | $-1.42$ | $-1.30$ | $-1.18$ | $+0.12$ |

system. The table also includes the well depth of the potential of mean force, $\Delta U_{\mathrm{PMF}}$, for each system obtained from the well-tempered metadynamics simulation at $T = 300$ K. Due to the inclusion of thermal motion and proper sampling of different cluster configurations, the values of $\Delta U_{\mathrm{PMF}}$ are typically less negative than the binding energies $\Delta E_{\mathrm{OPLS}}$ obtained using the same force field at $T = 0$ K. In the scope of the present study, we conclude that the OPLS all-atom force field provides an adequate representation of the individual collision partners as well as the formed clusters after collision.

*Author contributions.* IN, HV and BR planned the study. IN, RH and BR designed the simulation framework. IN carried out the ab initio calculations and IN and BR carried out the force field based molecular dynamics collision and well-tempered metadynamics simulations. RH provided the theoretical framework. IN, RH, and BR analyzed the simulation data, and wrote the first draft of the manuscript. All authors contributed to writing the final paper.

*Competing interests.* The authors declare that they have no conflict of interest.

*Acknowledgements.* This work was supported by the European Research Council (project 692891 DAMOCLES), Academy of Finland (grant nr. 337549) and University of Helsinki, Faculty of Science ATMATH project. Computational resources were provided by the CSC–IT Center for Science Ltd., Finland. The authors thank the Finnish Grid and Cloud Infrastructure (FGCI) for supporting this project with computational and data storage resources. The authors thank Valtteri Tikkanen and Huan Yang for stimulating discussions.



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
