# Peer review of "Modelling approaches for atmospheric ion-dipole collisions: all-atom trajectory simulations and central field methods"

_EGUsphere, 2022_

## Author Comment (AC1)

Dear editor and referee 1,

We sincerely thank referee 1 for their valuable feedback and corrections. Their comments helped us to recognize the areas where our manuscript could be improved. We took all comments to heart and have revised the manuscript accordingly. In the following, we provide point-by-point responses to general and specific comments. Referee comments are given in **_bold italic_**, while responses are given in roman (non-bold, non-italic). Excerpts from the revised manuscript to support our responses are written in yellow highlight. The line and page number to which a response refers to, is indicated by (L### P#).

We hope that the revisions in the manuscript and our accompanying responses prove sufficient, rendering our manuscript suitable for publication in _Atmospheric Chemistry and Physics_.

We look forward to hearing from you at your earliest convenience and thank you for considering out manuscript for publication.

Best regards,

Ivo Neefjes and Roope Halonen

**Referee 1 comments**

**General comments**

**_The present work describes molecular Dynamics (MD) simulations of collisions of molecules / clusters and ions and the determination of collision rate constants from these simulations as well as from theory, using analytical equations with different approximations. The study increases our understanding of the validity range and limitations of these approximations._**

We thank the referee for this general comment and their assessment that our study increases the understanding of the validity range and limits of the approximations used to determine collision rate coefficients.

**Specific comments**

**_Eq 11: to my understanding, $\beta_L$ is the rate constant for the collision of an ion with a polarizable molecule and $\beta_{SC}$ that for an ion with a polarizable dipolar molecule. Hence, I would expect that K should approach one for $\mu_d$ (and hence x) approaching 0, but it seems to be 0.025. Please comment on this behavior._**

We thank the referee for this observation. Eq 11. was indeed not properly reproduced in the manuscript. For the case $x \leq 2$, the correct equation for the correction term $K$ should be: $K = \frac{(x+0.5090)^2}{10.526} + 0.9754$. The last term ($+0.9754$) was missing in our manuscript. Equation 11 has now been corrected in the revised manuscript. As the study is limited to systems with $x > 2$, the erroneous equation given in the earlier version of the manuscript does not affect any of the presented results or discussion.

*I find the description of the equilibration process difficult to follow (lines 222-233). There is a statement "The collision partners were first separately equilibrated for 50 ps using a Langevin thermostat with a damping factor of 0.1 ps. During the equilibration, both the center-of-mass motion of each collision partner and the angular momentum of the total system were removed … Both collision partners were then given a velocity along the x-direction" If they are equilibrated separately (i.e., in separate simulations), both fragments should not have any rotational angular momentum. I have the impression rather "together but separated with a large distance" is meant. If the latter is the case, have you checked the correct distribution of rotational angular momenta? Only a check of the energy distribution is mentioned. This is a scalar, and the angular momentum is a vector, therefore this may not be sufficient.*

It is important to us that the description of the equilibration process is clear. The collision partners are indeed not equilibrated in separate simulations, but rather equilibrated at a distance beyond the cut-off for intermolecular interactions under separate thermostats. We only remove the very small spurious residual angular momentum of the combined system, which does not significantly affect the rotational motion of the individual collision partners and guarantees consistent starting conditions between individual collision trajectories.

The applied protocol assures random orientation of both partners in addition to canonical energy distributions for the rotational and vibrational modes of motion. According to our tests, each mode has on average $kT/2$ of kinetic energy which is in accordance with the equipartition theorem. Furthermore, the standard deviations of molecular temperatures for rotations and vibrations are also obeying the statistical prediction $T\sqrt{2/\mathrm{dof}}$, where dof is the number of available degrees of freedom. We strongly believe that in the context of this study, these tests and observations are sufficient to ensure that the thermodynamic state of the collision partners is well defined. We refrain from a more elaborate discussion of the equilibration here since we are preparing a separate manuscript specifically on this topic.

The description of the equilibration process has been clarified in the revised manuscript:

To obtain ion-dipole collision cross sections and rate coefficients from MD simulations, we determined the collision probability over a relevant range of impact parameters and relative velocities. All collision simulations were carried out with the LAMMPS code (Plimpton, 1995). At the start of the simulation, the collision partners were placed 600 Å apart along the $x$-axis, well beyond the cut-off of the Lennard-Jones and Coulomb potentials of the OPLS force field at 280 Å. At these positions, the collision partners were equilibrated for 50 ps using separate Langevin thermostats for each collision partner with a damping factor of 0.1 ps. During the equilibration, the center-of-mass motion of each collision partner was removed. We also removed the small spurious angular momentum of the combined system. A thorough analysis of different thermostats revealed that, for the studied flexible compounds, the Langevin thermostat is best suited to ensure equipartition of rotational and vibrational energies. Details of these investigations will be published elsewhere (Halonen et al. 2022).

(L219 P9)

*Line 261: "The center-of-mass distance criterion for a successful collision was determined for each system by taking the distance at which the value of the PMF was $5k_BT$ (~0.13 eV at 300 K) higher than its minimum," There are two such distances, right and left from the minimum. Which one have you chosen?*

We agree with the referee that the manuscript leaves an ambiguity here about which of the two distances is utilized in the analysis of the molecular dynamics results. We have now added the clarification "in the direction of increasing center-of-mass distance" to dispel any ambiguity in this statement.

The center-of-mass distance criterion for a successful collision was determined for each system by taking the distance at which the value of the PMF was $5k_BT$ (~0.13 eV at 300 K) higher than its minimum in the direction of increasing center-of-mass distance, to account for thermal fluctuations. (L264 P11)

***Line 270, Discarding and adding new trajectories in case of dissociation: Is that the correct procedure? To me, it would seem most appropriate to define 3 outcomes of the encounters: no collision, association, and dissociation of an existing dimer. Then one would use the total number of trajectories in the denominator for each of the rate calculations. This would seem in line with what is done in Master Equation calculations for barrierless reactions. Please comment on the justification for your approach.***

We appreciate the referee's concerns about our procedure of replacing trajectories if a dimeric collision partner dissociates. There are three reasons for why we have chosen this approach.

First, the aim of this study is to focus on the collision process independent from any other processes. By including dissociation into our analysis, we would effectively introduce a system-specific time scale and thus the rates would be affected by both the stability of the dimer and the chosen initial separation of the two collision partners. In the typical models of new particle formation, the association and dissociation processes are treated independently with specific rate coefficients (see e.g., McGrath et al. 2012). And the main aim of this study is to provide detailed rate coefficient for the collision process to be used in such models.

Second, the systems investigated in this work are characterized by very strong bonding, and subsequent dissociation of the initial dimer after collision with another partner was only observed for very few systems at the highest relative velocities and large impact parameters. To obtain adequate statistics of this process, we would have needed to increase the simulation time post collision as well as the number of individual trajectories. Given the rare occurrences and their extremely small contribution to the ensemble average when integrating over the Maxwell-Boltzmann distribution of relative velocities, we chose not to study this process further.

Finally, it is likely that the cluster dissociation process is unphysically enhanced due to the fully excited vibrational modes in the classical molecular model employed. In a quantized system at finite temperature, some high-frequency intramolecular vibrations possess no, or at least significantly less, energy than $k_B T/2$.

We hope that the referee accepts these justifications for our approach. We also kindly refer to the response to referee 2 for related discussions on the simulation setup.

***Line 279: What do you mean by "small oscillations in the interaction energy"?***

Due to their rotational motion, the collision partners are subjected to periodically oscillating electrostatic forces. Essentially, due to small, periodic repulsive forces at intermediate distances, the collision partners can slow down and eventually repel each other at small values of $v$ and $b$, which would otherwise be expected to yield certain collisions. This is demonstrated in our previous publication on the same topic (Halonen et al. 2019).

We have clarified this in the revised manuscript:

The reduced collision probabilities at low values of $v_0$ are caused by small oscillations in the interaction energy resulting from the rotational motion of the collision partners. These oscillations can lead to small, periodic repulsive forces at intermediate distances, causing the collision partners to slow down and eventually repel each other at small values of $v_0$ and $b$, which would otherwise be expected to yield certain collisions, as previously shown for dipole-dipole collisions (Halonen et al., 2019). (L283 P12)

***Can you specify statistical uncertainties, for the MD results, e.g., in table 2?***

For the MD simulations, we have estimated the errors for each $P(v, b)$ according to a binomial distribution: $\text{err}(v, b) = \sqrt{P(v, b)(1 - P(v, b))/N}$, where $N$ is the number of samples, here $N = 1000$. Thus, for simplicity, we have considered that the overall error for the collision rate coefficient arises only from the uncertainty given for collision cross section $\Omega_{\text{MD}}(v)$ (Eq. 15): The upper and lower limits of $\beta_{\text{MD}}$ are determined by calculating $\Omega_{\text{MD}}(v)$ for $P(v, b) \pm \text{err}(v, b)$. Since the statistical errors are consistently small (1.8 – 2.3 % for all systems), we haven't included them in Tab. 2 or any of the figures, instead we propose to add the following sentences in the manuscript text:

We determined upper and lower limits of $\beta_{\text{MD}}$ by calculating $\Omega_{\text{MD}}(v)$ in Eq. 15 for $P(v, b) \pm \text{err}(v, b)$. Here, $\text{err}(v, b)$ is estimated according to a binomial distribution $\text{err}(v, b) = \sqrt{P(v, b)(1 - P(v, b))/N}$, where $N = 1000$ is the number of samples. Due to the large number of samples, the estimated error is on average only 2%, with the largest error being 2.3% for the $H_2SO_4 - [HSO_4^- \cdot H_2SO_4]$ system. (L311 P14)

**Technical comment**

***Figs 4 and 5: The Circles do not show up on my Adobe Acrobat on android, but they do on windows.***

We thank the referee for checking the integrity of the figures with multiple PDF viewers. We will ensure the vector PDF graphics are as "portable" as possible in the final accepted version of the manuscript.

[Figure]

[Figure]

Dear editor and referee 2,

We sincerely thank referee 2 for their valuable feedback and comments. We have taken their suggested improvements to heart and have revised the manuscript accordingly. In the following, we provide point-by-point responses to the referee's comments. Referee comments are given in **_bold italic_**, while responses are given in roman (non-bold, non-italic). Excerpts from the revised manuscript to support our responses are written in yellow highlight. The line and page number to which a response refers to, is indicated by (L### P#).

We hope that the revisions in the manuscript and our accompanying responses prove sufficient, rendering our manuscript suitable for publication in *Atmospheric Chemistry and Physics*.

We look forward to hearing from you at your earliest convenience and thank you for considering out manuscript for publication.

Best regards,

Ivo Neefjes and Roope Halonen

**Referee 2 comments**

***This manuscript explores the collision dynamics of eight ion-dipole systems using potential of mean force (PMF) calculations and molecular dynamics (MD) simulations. Collision probability maps are obtained by MD to determine the dynamic collision cross sections. The collision rate coefficient results obtained by PFM and MD are compared to the classic Su and Chesnavich and the Langevin-Gioumousis-Stevenson models.***

***The manuscript is in general well written and well organized and manages to provide an understanding of the conditions that PMF calculations and central field models can be used reliably for the determination of the collision rate coefficient.***

We are very grateful to the referee for their kind comments. We have gone through various drafts of the manuscripts before arriving at the current structure. It is nice to hear that the result of this work is being appreciated.

***To further improve the manuscript, I suggest adding a comment on how long the clusters are traced after collision, i.e., what is the lifetime of the clusters shown here.***

In the present study, the focus lies on the statistics of collisions. To perform the simulations efficiently, we only ensure that the simulation time is sufficient to capture a potential collision. Any subsequent processes, such as dissociation or evaporation, which may occur on much longer timescales, are disregarded.

The number of timesteps for each molecular dynamics simulation was determined by the time it takes for two non-interacting collision partners to cross each other on the x-axis (distance along x-axis / relative velocity) plus 50 ps. Due to the interactions between the collision partners and the impact parameter, the actual time it takes for a collision to occur can be slightly shorter or longer than the non-interacting case. In general, we can, however, say that we trace formed clusters for around 50 ps after collision.

We kindly refer to the response to referee 1 for additional discussions on the topic of dimer dissociations after collision.

To make this point clearer, we have added the following revision to the manuscript:

The duration of the simulation was dependent on the initial relative velocity. It was determined as the time it would take for two non-interacting particles to cross each other plus 50 ps, to ensure all potential collisions are captured. This simulation procedure is not meant to study any post-collision processes, such as dissociation or evaporation, which can occur on longer timescales, involving redistribution of energy and thermalization of the formed cluster. (L238 P9)

***In addition, the value of the probability, P(v,b), at which the dynamic collision cross sections of Fig. 4(e-h) and Fig. 5(e-h) were obtained by MD, should be provided.***

In the MD collision simulations, the dynamical collision cross sections reported in Figs. 4 and 5 were obtained from the integral over the collision probabilities over the range of impact parameters simulated according to Eq. (15),

$$\Omega_{MD}(v) = \pi \int_0^\infty db^2 P(v, b).$$

To make this clearer, we have modified the corresponding descriptions of panels (e-h) in the captions of Figs. 4 and 5:

Corresponding dynamic collision cross sections $\Omega_{MD}(v)$ obtained from these MD collision probabilities using Eq. (15) (open circles) and collision cross sections $\Omega_{CF}(v)$ based on the central field model (solid lines) using an attractive interaction fitted to the PMF according to Eq. (5) (e–h, bottom row). (Fig. 4 P12 & Fig. 5 P13)

**Additional changes to the manuscript:**

Upon careful examination of the manuscript, we noticed a spurious factor of $\pi$ in Eq. (16) which has been removed in the revised manuscript:

$$\beta_{MD} = \int_0^\infty dv \, v \, f_{MB}(v) \, \Omega_{MD}(v). \tag{16}$$

(Eq. 16 P14)